# Analysis of the Influencing Factors of Organizational Resilience in the ISM Framework: An Exploratory Study Based on Multiple Cases

Yingqi Liu [1], Ruijun Chen [1,2], Fei Zhou [1,*], Shuang Zhang [1] and Juan Wang [3]

1   School of Economics and Management, Beijing Jiaotong University, Beijing 100044, China; Liuyq@bjtu.edu.cn (Y.L.); 18113079@bjtu.edu.cn (R.C.); 21120714@bjtu.edu.cn (S.Z.)
2   Department of International Economics, Government and Business, Copenhagen Business School, 2000 Copenhagen, Denmark
3   School of Business Administration, Fujian Business University, Fuzhou 350108, China; 18120715@bjtu.edu.cn
*   Correspondence: 19113079@bjtu.edu.cn

**Abstract:** As an important means to deal with crisis, organizational resilience has attracted the attention of academia and industry. However, research on what factors influence organizational resilience has lagged behind. In view of this, this study proposes the concept of organizational resilience on the basis of existing research and extracts the influencing factors of organizational resilience based on a multi-case analysis approach, using the organizational behavior of five companies in crisis situations as the research object. Based on the Interpretive Structure Model (ISM), the internal logical relationship and hierarchical structure of the factors influencing organizational resilience are analyzed. In this study, the importance of influencing factors of organizational resilience was analyzed by using analytic network process (ANP). It is suggested that strengthening organizational resilience is the key, organizational learning is the important basis, emotion management is the necessary condition, and organizational resources are the basic guarantee, which provides theoretical supplement and practical guidance for the study of organizational resilience.

**Keywords:** organizational resilience; influencing factors; ISM; ANP; multi-case study

## 1. Introduction

In the context of economic globalization, companies are facing an unprecedented uncertain business environment in which unexpected events are ubiquitous [1], such as the Indian Ocean tsunami in 2004, the Chilean earthquake in 2010, and the recent novel coronavirus (COVID-19) [2]. Natural disasters, pandemic diseases, terrorist attacks, political unrest and economic instability all have unpredictable effects on organizational sustainability and competitiveness [3]. Some crises may provide opportunities for companies to grow [4], such as building new organizational relationships [5]. However, crises more often catch organizations off guard, create uncertainty for members in the organization [6], and even lead to organizational disintegration [7–9]. Facing these crises makes us reflect on how organizations can profit from the turbulent environment and achieve sustainable development and gain competitive advantage [10]. The history of Southwest Airlines seems to give us a good insight. The events of 11 September 2001 had a disastrous impact on the U.S. airline industry, but Southwest achieved profitability in 2011 and maintained its record of continuous profitability through 2017. Thus, resilient organizations are able to thrive after a crisis [11]. Empirical and theoretical studies have shown that organizational resilience can explain how organizations survive and thrive in the face of adversity or turbulence [12]. Organizational resilience not only helps organizations live longer by improving their ability to withstand and adapt to environmental changes [13,14], but also enables organizations to maintain a long-term competitive advantage [15]. Therefore,

today more than ever, companies need to pay extra attention to fostering organizational resilience [16] and tapping into the core influences of organizational resilience is the key to finding out how organizations can achieve sustained competitive advantage.

Over the past decade, organizational resilience has received increasing attention in academic and theoretical circles [17–21]. Positive psychology [22], engineering [23], ecology [24,25], management [20,26] and other fields have been discussed extensively and the related literature is climbing year by year [27]. However, to date, there is no clear and unambiguous interpretation of the relationships among the factors influencing organizational resilience.

Existing studies have focused on examining the influence effects of individual factors on the one hand, and the findings are fragmented. For example, Mafabi et al. (2013) [28] pointed out that an organizational culture that supports innovation and openness is a key factor in organizational resilience. In such an environment, employees can be encouraged to share their perceived real-time information about potential problems that the organization may encounter in the future. Further, employees can only activate personal resilience in an organizational environment that supports or actively promotes resilient organizational behavior [15]. Without environmental support, there is no way to translate resilience perceptions and behaviors into organizational capabilities [11]. Andersson et al. (2019) [29] concluded that balancing organizational structure to enhance organizational resilience through a longitudinal qualitative case study of the Swedish Bank. On the other hand, the findings of different scholars on the influencing factors are inconsistent. For example, the relationship between organizational learning and organizational resilience. Chabot (2008) [30] found that training is a key element to enhance organizational viability. Therefore, organizational learning helps to enhance organizational resilience. Similarly, Mithani et al. (2021) [31] stated that higher learning capacity is beneficial to enhance the speed of organizational recovery after a crisis. Vogus and Sutcliffe (2007) [32] studied that organizational learning is both an input and a result of organizational resilience. The discrepancy in the findings suggests that further research and exploration of the factors influencing organizational resilience is still needed.

At present, in terms of research area, Chinese scholars are relatively scarce in their research on the influencing factors of organizational resilience, and only a systematic review of existing research on organizational resilience has been conducted. In terms of research content, foreign researchers on the influencing factors of organizational resilience mainly focus on quantitative research, focusing on the effect of single influencing factors on organizational resilience. There are few multi-case studies based on qualitative data, and the results are relatively scattered. In view of this, this study extracts the influencing factors of organizational resilience through multiple case studies of Southwest Airlines, Starbucks, Lego, Apple, and Kyocera. Then, based on the perspective of system analysis, the obtained data are analyzed with the Delphi method, and the key indicators of the factors affecting organizational resilience are selected. In order to provide theoretical and empirical guidance for enterprises to better improve organizational resilience, we use the ISM to deeply analyze the internal logical relationship among the influencing factors and the hierarchical structure among the influencing factors. Finally, the results obtained from ISM are then used as input variables to calculate and rank the weights of each influencing factor using ANP, so as to find the main influencing factor. It also helps enterprises to enhance organizational resilience and provides theoretical and empirical guidance for the survival and development of enterprises.

Second, the existing exploration of the influencing factors of organizational resilience is fragmented. This study systematically organizes the influencing factors of organizational resilience from three levels, namely surface, middle and deep levels, based on the studies of Valero et al. (2015) [33], Amir (2018) [34], Mithani et al. (2020) [31] and others. From a system perspective, a multi-case research analysis method is used to derive the influencing factors of organizational resilience. On this basis, the ISM of organizational resilience was constructed to deeply explore the inner logical relationship among the influencing

factors of organizational resilience and the hierarchical structure among its influencing factors, which provides theoretical guidance for the subsequent exploration of the path of organizational resilience.

Third, this study analyzed the importance of organizational resilience influencing factors using ANP. It found that the main factors affecting organizational resilience are organizational resources, organizational capabilities, organizational relationships, organizational communication, social capital, organizational strategy, organizational learning, and work passion. It was also found that organizational resources and organizational capabilities are the two most important factors influencing organizational resilience, which is consistent with the resource-competency doctrine in existing studies.

This paper is organized as follows: the second part reviews the connotation and influencing factors of organizational resilience and analyzes the influencing factors. The third part conducts a multi-case analysis of the influencing factors of organizational resilience, describes the criteria for case selection, data collection and analysis strategies, and the process of data analysis. The fourth part uses the ISM approach to analyze the influencing factors of organizational resilience and constructs an ISM of the influencing factors of organizational resilience. The fifth part analyzes the importance of the influencing factors of organizational resilience by constructing an ANP model. The sixth part concludes and discusses the results obtained from this study and presents the future research outlook.

## 2. Literature Review

### 2.1. The Connotation of Organizational Resilience

The term resilience originates from the field of physics and refers to the ability of a material to return to its original form after deformation. It is also used to describe the ability of a system to absorb changes and still maintain its basic function [35]. Holling (1973) [36] first introduced resilience to social ecology in his article "Resilience and stability of ecological systems" and argued that resilience is closely related to the stability of ecosystems. Subsequently, resilience was gradually introduced into ecology, economics, psychology, and sociology, to describe key features of complex dynamic systems. As research progressed, Wildavsky (1988) [37] first included resilience in the study of organizations. However, it was not until the late 1990s that the study of resilience in organizations gradually gained favor among scholars, who began to focus on the study of resilience after disasters [2,29,38]. In organizational research, the concept of resilience has been applied in crisis management, disaster, and high reliability organizational literature [39,40]. In recent years, organizational resilience has been well developed in the field of psychology, where researchers have argued that organizational resilience is the positive adaptive capacity that organizations exhibit when experiencing adverse conditions by using children in high-risk situations as subjects [40].

In addition, with the changing of research objects and purposes, the connotation of resilience has been given different meanings. The disagreement among scholars is the "stability" and "equilibrium" emphasized by traditional resilience and the "evolutionary" and "non-equilibrium" shown by resilience in reality. Therefore, with the change of the research field, the connotation and characteristics of resilience will also change. In this paper, the connotation of resilience is explored more clearly through the summary of engineering resilience, adaptive resilience, and ecological resilience in terms of connotation, applicable objects, and characteristics, as shown in Table 1 below.

As can be seen from Table 1, with the change of the research object, the concept of resilience has been improved. There is a gradual shift from the previous balanced, static, and stable, to unbalanced, dynamic, and diverse. The concept of adaptive resilience differs from engineering resilience and ecological resilience in that it applies to both organizational and economic systems. It places more emphasis on the adaptive capacity that different elements within the organizational system possess or embody in response to a crisis. For example, Seville et al. (2008) [41], through a study of New Zealand organizations that

have endured disasters, viewed resilience in terms of organizational capacity to improve the effectiveness of organizations in managing risk through adaptive capacity.

At present, academic circles have not formed a unified conclusion about what is organizational resilience [7,42]. By summarizing and distilling the literature it is clear that organizational resilience is a multidimensional, cross-level, and complex concept [40]. There are two main views on the concept of organizational resilience. One is the scholars who hold the "dynamic view". They believe that organizational resilience is a dynamic capability or development process that can be developed, and they advocate defining organizational resilience from the perspective of capability and process. Second, scholars with a "static view" regard organizational resilience as an ideal trait possessed by an organization or a coping result realized, and advocate defining organizational resilience from a functional perspective and an outcome perspective. Based on the process perspective, scholars believe that organizational resilience is a dynamic evolutionary process. In this process, the organization adjusts its configuration to cope with the external adverse environment, which may involve reintegration, improvisation, resource allocation, emotional labor, etc. [43,44]. From the perspective of capability, scholars believe that organizational resilience is a dynamic and flexible organizational capability, which is composed of many abilities. These include stability maintenance ability, endurance ability, coping ability, development ability, learning ability, prediction ability, and survival ability displayed by an organization in a crisis situation [1,11,45]. Based on the outcome perspective, scholars believe that organizational resilience is the result of organizations maintaining good adaptability in the face of adversity, and it is related to how organizations recover and survive in chaotic changes and unexpected events [46,47]. From a functional perspective, scholars believe that organizational resilience is a function of an organization's understanding of the overall situation, management of key weaknesses, and ability to adapt in a complex, dynamic, and interdependent environment [48,49].

**Table 1.** Connotation and characteristics of different resilience concepts.

| Category of Resilience | Connotation | Characteristics | Applicable Objects |
|---|---|---|---|
| Engineering Resilience | The ability of a system to recover or return to its original state after a shock or disturbance. The concept emphasizes the equilibrium stability of the system state after a response (Walker, 2006) [50]. | Recoverability, single equilibrium, static stability | Physical System and Engineering Systems |
| Ecological Resilience | The possibility of the system developing to another state after a disturbance (which may be lower than the original equilibrium state, may decline, or may move to a better state), it emphasizes the multiple stability of the system (Simmie and Martin, 2010) [51]. | Intermittent equilibrium, multiple equilibria, dynamic stability | Ecosystem |
| Adaptive Resilience | Systems minimize the impact of shocks or disturbances by mutual adaptation and co-evolution after being subjected to a shock or disturbance. The concept emphasizes the adaptive capacity of the system (Martin, 2012) [52]. | Complex adaptation, non-equilibrium, dynamic evolution | Organizational System and Economic System |

Source: Compiled by the author.

*2.2. Influencing Factors of Organizational Resilience*

From the research within the study, only scholars Vakilzadeh and Haase (2021) [53] summarized and explored the influencing factors of organizational resilience based on the empirical study of organizational resilience; however, it is difficult to reflect the cross-level characteristics of organizational resilience. In view of this, this study systematically sorts out the influencing factors of organizational resilience from three levels, namely

surface, middle and deep levels, based on the previous studies, in order to more clearly reflect the characteristics of the influencing factors of organizational resilience.

For the surface-level influences, i.e., which influences directly affect organizational resilience, the main ones include organizational capacity, organizational relationships, organizational learning, and organizational communication. Capacity is an aspect of an organization that is necessary to perform its functions and achieve its core mission and vision. An organization has a greater ability to respond to crises facing the organization if it has a stable structure and the right configuration of people, funding, technology, and decapitation plans [54]. Valero et al. (2015) [33] found through a study of public and nonprofit organizations that financial and people capabilities, which are included in organizational capacity, have a positive effect on organizational resilience. Numerous studies have shown that positive interpersonal relationships improve individual, community, and organizational outcomes [55,56], and can make it easier for organizations to overcome difficulties and resume operations. Gittell et al. (2006) [9] found through a study of 10 airlines after the September 11 incident that positive relationships or relationship reserves at work are a prerequisite for organizational resilience. They found that the company's decision not to lay off employees, positive internal relationships, adequate financial reserves, and a viable business model all contributed to organizational recovery from the crisis. They argue that positive relationships play an important role in explaining organizational resilience. Positive relationships tend to result in lower costs and lower debt levels over time, making it easier to respond to external shocks without breaking commitments, further strengthening relationships and performance. Regarding the organizational learning dimension, Chabot (2008) [30] states that training as a key element of organizational viability and organizational learning can contribute to the understanding of organizational resilience. Mithani et al. (2021) [31] found that higher learning capacity increases the speed of organizational recovery after a threat occurs and that having slack and learning capacity helps to ensure organizational resilience. Regarding organizational communication, researchers have found that resilience depends on the ability of affected parties to communicate and organize during periods of rapid change or disruption. It encompasses the ability of firms to respond to crises and adapt in creating new solutions [5]. In addition, communication is also seen as an important factor in shaping organizational resilience. Organizational communication helps to achieve shared situational awareness, as well as better interpretation and assessment of critical situations, resulting in more consistent and reliable decision-making processes in such situations [1,57]. Thus, Lengnick-Hall et al. (2011) [11] creatively identified open communication and collaboration as important methods to promote organizational resilience.

Mid-level influences are factors that need to be reflected by a deeper inquiry into organizational behavior. The main ones include organizational culture, organizational structure, and organizational leadership. For organizational culture, an organizational culture that supports innovation and is open is considered a key factor for organizational resilience [15,28]. Teixeira and Werther (2015) [15] state that an environment of openness and trust is a key element of organizational resilience. This environment helps to encourage employees to share real-time information about potential problems they perceive the organization may encounter in the future. Furthermore, employees are better able to activate personal resilience only in an environment that supports or actively promotes organizational resilience behaviors. Without the support of the environment, the perception and behavior of resilience cannot be translated into organizational capabilities [11]. For organizational structure, Andersson et al. (2019) [29] concluded balancing the organization (corporate philosophy, decentralized structure, information systems, and human resource management processes) to enhance organizational resilience through a longitudinal qualitative case study of a Swedish bank. In addition, balancing power distribution (achieved through decentralized structures and use of information systems) and normative control (achieved through organizational philosophy and human resource management processes) is important to achieve organizational resilience. For organizational leadership, Teo et al.

(2017) [58] used a case study of a hospital in Singapore during the SARS crisis as an example to show that leadership is critical to enhance organizational resilience in a crisis. They developed the RAR model to elucidate the leader's activation of organizational recovery through the cognitive, social, and emotional reserves inherent in the social network through the lens of relational networks. Furthermore, Teixeira and Werther (2015) [15] state that leadership is a key factor in building resilient organizations. The essence of leadership in organizations lies in the process of facilitating individual and collective efforts to achieve common goals [48]. Leadership of leaders occurs throughout the different phases of the crisis period in which the organization is in (pre-crisis, at the height of the crisis, or during recovery) and has different impacts on the organization [59].

Deep influence factors, which are the factors that affect organizational resilience at the deepest level, mainly include social capital, organizational resources, cognitive ability, and emotional ability. Social capital, as a collection of actual or potential resources embedded in a persistent institutional or systemic social network, has been shown to have a significant impact on organizational resilience as an environmental factor. Among them, social capital within the organization helps to improve the quality and effect of knowledge transfer among the members of the organization [60,61]. It has also been found that intra-organizational social capital affects the degree of coordination and cooperation in employees' work [57,62]. Therefore, to a certain extent, it can affect employees' productivity and work motivation, thus enhancing organizational resilience in the face of crises. In addition, sociologists who advocate resource dependence emphasize that expanding resource networks is a key factor in creating resilience in organizations, and such individuals in organizations will achieve better performance by maintaining good interpersonal relationships with colleagues who have key information resources [63]. Meanwhile, many studies have pointed out that resource availability is considered a key driver of organizational resilience [64–66]. The depletion of organizational resources can severely limit their ability to recover from shocks. This is because "sufficient internal resources and the ability to rearrange, transform, and adapt these resources to uncertainty and changing post-shock economic conditions" are key elements of organizational resilience flexibility [67]. As Pal et al. (2014) [66] observed, resource constraints, especially physical, financial and technological, weakened the resilience of Swedish SMEs in response to the economic crisis. In addition, good cognitive ability represents a clear sense of vision and purpose, a firm sense of value and professional knowledge [68,69], which allows flexible and efficient feedback in the face of unexpected events. Positive emotional competencies include optimism, hope, and having the opportunity to express and discuss emotional opportunities [68,70,71], which can identify more opportunities to reduce losses and achieve better stability in a crisis.

Thus, the literature has enriched the theoretical study of organizational resilience. However, the analysis of the literature reveals that there is a lack of research on organizational resilience in China. In terms of the content of the research, the question "What factors affect organizational resilience?" and "What is the interrelationship between the influencing factors?" and other related studies are rare, and systematic studies on the factors affecting organizational resilience are still insufficient.

## 3. Multi-Case Analysis of Factors Influencing Organizational Resilience

Case study methods can be used to study the evolution process of behavior development by mining and analyzing qualitative data [72] combined with the actual situation. Therefore, based on the study of the connotation and influencing factors of organizational resilience, this study selects cases of typical enterprises to extract the influencing factors of organizational resilience from the effect of organizational resilience.

### 3.1. Case Selection

To further explore the factors influencing organizational resilience, this study selects cases based on the principles of theoretical sampling, taking into account the typicality of

the cases drawn and the ease of access to information. In this regard, theoretical sampling refers to the selection of the sample that best illustrates the research question, provided that the research question and the direction of the study are relatively clear [68]. Theoretical sampling differs from random and stratified sampling in that the samples selected are not intended to test established theoretical hypotheses but to construct and develop new theoretical explanations. Rather than focusing on the number of samples, theoretical sampling selects data sources that are most closely related to theoretical constructs. The data are collected and analyzed, and the categories and concepts reflected in the case sample are continuously extracted until a phenomenon can be explained and the relationship between categories or the interaction between concepts has sufficiently reached a state of theoretical saturation, meaning that theoretical sampling is complete [73]. In addition, taking into account the typicality of the cases, this study selects companies with a development history of more than 40 years, which have encountered major crises and have successfully emerged from them to achieve sustained growth. At the same time, this study selects companies that can obtain information from news reports, published books, and industry materials, taking into account the ease of access to information. Based on these considerations, Southwest Airlines, Apple, Microsoft, Starbucks, and Kyocera were selected as samples for this study.

### 3.2. Data Collection and Analysis Strategy

In this study, news reports, published books, and industry information about the resilience of organizations, such as Southwest Airlines, Apple, Microsoft, Starbucks, and Kyocera, were used as important data sources. The authenticity and validity of the materials were emphasized in the selection and sources of materials. At the same time, experts and doctoral students in this field were invited to analyze the obtained information, and the analysis results were compared to ensure the accuracy of the data analysis. Accordingly, a database of more than 200,000 words was constructed, which laid the foundation for the smooth implementation of the case study.

### 3.3. Multi-Case Factor Extraction

On the basis of collating and summarizing the obtained data and combining with previous studies, this study finally summarizes 20 key factors that affect the improvement of organizational toughness, as shown in Table 2 below. Meanwhile, this study summarizes the 20 influencing factors into five aspects, among which, organizational communication, organizational learning, organizational commitment, organizational change, and organizational efficiency are organizational action factors. Organizational strategy, business model and organizational leadership are organizational model factors. Organizational structure, organizational culture and social responsibility are organizational attribute factors. Organizational competence, emotional competence, cognitive competence, threat perception and work passion are organizational competence factors. Organizational resources, organizational relationships, social capital, and organizational trust are organizational resource factors. All of these are shown in Figure 1.

**Table 2.** Table of multi-case analysis of factors influencing organizational resilience (partial).

| Example of Original Statement | Conceptualization | Categoryization (Extraction Factors) |
| --- | --- | --- |
| Our goal is to design a capital structure that leverages various capital levers to maximize returns for our shareholders over the long term. We will combine the strengths and weaknesses of our people to move them to positions that will allow them to perform their duties better. We have created a flat organizational structure to allow for quick and effective communication between organizations and departments. | Capital Structure Personnel Structure Hierarchy | Organizational Structure |
| Usually when we encounter a problem that cannot be solved, effective communication with our superiors is the most effective way. Sometimes effective communication between organization members will help us to eliminate misunderstandings, better identify and solve problems, and make the departments more coordinated and work in tandem. When there is a major crisis, we increase the number of communications to help us respond in a timely manner to the crisis that arises. | Communication with superiors Horizontal Communication Frequency of communication | Organizational Communication |
| Some issues require specialized PR staff to deal with, and recruit these people in case they are needed. When a business is in crisis, tightening financial resources can help us get through the crisis and help us rise again in the future. There was a time when everyone was experiencing a crisis and pooling the most advantageous resources, or even making resource substitutions, became the source of everyone living and even competing. | Human Resources Financial Resources Material Resources | Organizational Resources |
| Effective decision-making by leaders helps us find our way through confusing decisions. At the same time, we sometimes have a lot of hesitation, and the leader's typography helps us better determine our direction. We all actively encourage our employees to be bold and innovative, and the example and inspiration effect of leaders is the source of passion for employees. Leaders also need to continue to learn so that they can better understand the problems that the business may encounter in the future. | Leadership decisions Leadership Inspiration Leadership Learning | Organizational leadership |
| We often reflect dialectically on each step we take, profoundly reflecting on the accuracy and timeliness of our responses, and continuously optimizing our decisions in our reflections. Learning from other companies is an effective way to deal with the crisis, and we will obtain relevant experience from related companies to make up for our shortcomings. Sharing knowledge and experience with members of our organization and other organizations helps us to better communicate, enrich our experience and enhance our capabilities. | Critical Reflection Organizational Acquisition Organizational Sharing | Organizational Learning |

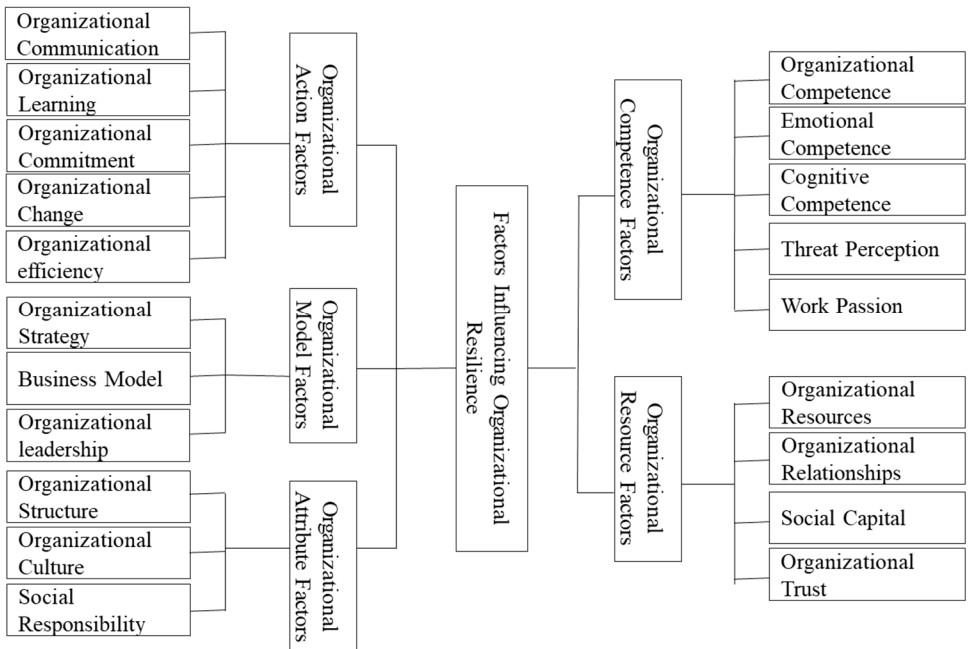

**Figure 1.** Index system of factors influencing organizational resilience.

## 4. Research Methodology and Process

### 4.1. Research Methodology

The Interpretative Structure Model (ISM) is an analytical method in systems engineering theory. It is designed to decompose the set of factors affecting the complex system into several sub-elements and find out the relationship between each element and finally form the structure relationship matrix diagram. The method was proposed by American systems engineering theorist Professor Warfield in 1973 to transform ambiguous ideas and views into intuitively clear and well-structured models. It emphasizes that the analysis of things needs to be rooted in collected realistic materials and the processing and analysis of the information. Through theoretical deduction, the interaction mechanism among various combination elements in the complex system is extracted, and the theoretical construct is finally formed. Compared with other empirical analyses of influencing factors, the main feature of the data collection method of the ISM is that it can be continuously supplemented with the required data according to the dynamics of research progress. In this way, the richness, tightness, and saturation of the information data can be ensured, and the persuasion of the research conclusions can be enhanced.

The basic idea of the theoretical method is to determine the research topic through problem analysis, with the help of a variety of creative techniques, to extract the impact (cause) factors of the problem. Through the design of influencing factors, relations, such as wizard diagram, structure matrix, statistical software, and other technical tools to process the information of influencing factors and their relationship, finally form a multi-level hierarchical interpretive structure system conceptual model. In order to improve the knowledge and understanding of the conceptual model, it is necessary to test and repair the model with the help of practical cases for theoretical saturation, as shown in Figure 2. The advantage of this theoretical approach is that it can clarify the combination elements and their interrelationships in complex systems so as to facilitate understanding and control.

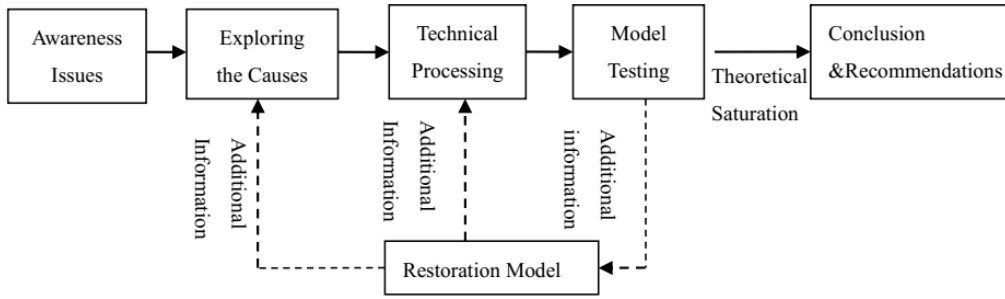

**Figure 2.** Flowchart of the ISM study.

*4.2. Model Building and Calculation Process*

Based on the influencing factors obtained by combining domestic and foreign studies and case studies, this study uses the Delphi method and solicits opinions from experts, scholars, and entrepreneurs in the field with the help of emails and interviews, and finally identifies 20 influencing factors, which are coded and interpreted as shown in Table 3 below.

**Table 3.** Factors influencing continuous entrepreneurial action.

| Influencing Factors | Code | Description |
|---|---|---|
| Organizational Structure | $F_1$ | Organizational structure refers to how work tasks are divided, grouped and coordinated for cooperation. |
| Organizational Communication | $F_2$ | Organizational communication refers to the exchange and transfer of information within an organization. These include a wide range of information, such as news, information, knowledge, experience, etc. |
| Organizational Resources | $F_3$ | Organizational resources are the indicators of resources and capabilities at the overall level of the enterprise, the application and integration of individual resources, mainly in the corporate culture and spirituality, corporate image and reputation, the organization's coordination ability, learning ability and resilience. |
| Organizational leadership | $F_4$ | Organizational leadership is is an operational process that brings a group of people together to function according to set goals. |
| Organizational Learning | $F_5$ | Organizational learning refers to the various actions taken by an organization around information and knowledge skills in order to achieve development goals and improve core competencies; it is the process by which an organization continuously strives to change or redesign itself to adapt to a continuously changing environment. |
| Organizational Relationships | $F_6$ | Organizational relationships refer to the status and interrelationship of the organization's personnel, such as the organization's institutional set-up and the division of management authority. |
| Organizational Competence | $F_7$ | Organizational capability refers to the ability to carry out organizational work and is the ability of a company to transform its various factor inputs into products or services with the same level of productivity or higher quality as its competitors' inputs. |
| Organizational Trust | $F_8$ | Organizational trust refers to the emotional confidence and support that employees hold in their hearts for the organization. |
| Organizational Strategy | $F_9$ | Organizational strategy refers to the planning and decision making of the organization regarding the overall, long-term and programmatic goals. It is the planning and decision-making of the organization on the global, long-term and programmatic goals of production and management and sustainable and stable development in order to adapt to the changes in the future environment. |

**Table 3.** *Cont.*

| Influencing Factors | Code | Description |
|---|---|---|
| Business Model | $F_{10}$ | The business model refers to the various trading relationships and connections between companies, between departments of companies, and even with customers and channels. |
| Emotional competence | $F_{11}$ | Emotional competence means that happiness and sadness are normal human reactions, and that emotions flow naturally in moderation. |
| Social Responsibility | $F_{12}$ | Social responsibility refers to an organization's responsibility to society. |
| Social Capital | $F_{13}$ | Social capital refers to the associations between individuals or groups—social networks, norms of reciprocity and the resulting trust—and is the resource that people bring to their position in the social structure. |
| Threat Perception | $F_{14}$ | Threat perception refers to an organization's ability to perceive threats that arise from outside sources. |
| Work Passion | $F_{15}$ | Work passion is defined as a strong tendency of people willing to invest time and energy in their work, with good explanatory power for burnout, creativity, happiness, performance, etc. in the workplace. |
| Organizational Commitment | $F_{16}$ | Organizational commitment is the identification with and trust in the goals and values of the organization to which an individual belongs, and the positive emotional experiences that result. |
| Organizational Efficiency | $F_{17}$ | Organizational efficiency refers to the proportional relationship between the output of social organizations of all levels and types and their managers engaged in management activities and the human, material and financial resources consumed, and is the concrete embodiment of management functions. |
| Organizational Change | $F_{18}$ | Organizational change is the process of adjusting, improving and innovating elements of an organization (such as its management philosophy, work style, organizational structure, staffing, organizational culture and technology, etc.) in a timely manner in response to changes in the internal and external environment. |
| Organizational Culture | $F_{19}$ | Organizational culture refers to an organization's unique cultural image consisting of its values, beliefs, rituals, symbols, ways of doing things, etc. Simply put, it is the various aspects of a company that are expressed in its daily operations. |
| Cognitive Competence | $F_{20}$ | Cognitive ability refers to the human brain's ability to process, store and extract information, that is, people's ability to grasp the composition of things, the relationship between performance and other things, the dynamics of development, the direction of development and basic laws. |

### 4.2.1. Build the Adjacency Matrix

In this study, a $20 \times 20$ matrix is used to represent the logical relationship between the factors influencing organizational toughness, which leads to the adjacency matrix A. The element $a_{ij}$ in the adjacency matrix represents the element in row i and column j, i.e., it represents the correlation between the factors influencing organizational resilience $F_i$ and $F_j$. Where, i, j = 1, 2, ......, 20. The adjacency matrix A is represented as follows.

$A = [a_{ij}]_{20 \times 20}$, where:

$$\text{When } a_{ij} = \begin{cases} 1, \text{ Means that element } F_i \text{ has a direct effect on } F_j \\ 0, \text{ Means that elements } F_i \text{ have no direct effect on } F_j \end{cases}$$

In order to ensure the scientific validity of the analysis results, the adjacency matrix was established by using expert consultation and a brainstorming method. The questionnaire was prepared and sent to experts, scholars, and entrepreneurs in the field of

the organization. The relationship among the 20 influencing factors above was compared, filtered, and selected. Finally, the opinions of most experts were adopted to obtain the adjacent matrix A, as shown in Figure 3 below.

| | 1 | 2 | 3 | 4 | 5 | 6 | 7 | 8 | 9 | 10 | 11 | 12 | 13 | 14 | 15 | 16 | 17 | 18 | 19 | 20 |
|---|---|---|---|---|---|---|---|---|---|---|---|---|---|---|---|---|---|---|---|---|
| $F_1$ | 0 | 1 | 0 | 0 | 0 | 0 | 0 | 0 | 0 | 0 | 0 | 0 | 0 | 0 | 0 | 0 | 0 | 0 | 0 | 0 |
| $F_2$ | 0 | 0 | 0 | 0 | 0 | 1 | 0 | 0 | 0 | 0 | 0 | 0 | 0 | 0 | 0 | 0 | 0 | 0 | 0 | 0 |
| $F_3$ | 0 | 0 | 0 | 0 | 0 | 0 | 0 | 0 | 1 | 0 | 0 | 0 | 1 | 0 | 0 | 0 | 0 | 0 | 0 | 0 |
| $F_4$ | 0 | 1 | 0 | 0 | 0 | 0 | 0 | 0 | 0 | 0 | 0 | 0 | 0 | 0 | 0 | 0 | 0 | 0 | 0 | 0 |
| $F_5$ | 0 | 0 | 0 | 0 | 0 | 0 | 1 | 0 | 0 | 0 | 0 | 0 | 0 | 0 | 0 | 0 | 0 | 0 | 0 | 0 |
| $F_6$ | 0 | 0 | 0 | 0 | 0 | 0 | 0 | 0 | 0 | 0 | 0 | 0 | 0 | 0 | 0 | 1 | 0 | 0 | 0 | 0 |
| $F_7$ | 0 | 0 | 0 | 0 | 0 | 0 | 0 | 0 | 0 | 0 | 0 | 0 | 0 | 0 | 0 | 0 | 1 | 0 | 0 | 0 |
| $F_8$ | 0 | 0 | 0 | 0 | 0 | 1 | 0 | 0 | 0 | 0 | 0 | 0 | 0 | 0 | 0 | 1 | 0 | 0 | 0 | 0 |
| $F_9$ | 0 | 0 | 0 | 0 | 0 | 0 | 0 | 0 | 0 | 1 | 0 | 0 | 0 | 0 | 0 | 0 | 0 | 1 | 0 | 0 |
| $F_{10}$ | 0 | 0 | 0 | 0 | 0 | 0 | 0 | 0 | 0 | 0 | 0 | 0 | 0 | 0 | 0 | 0 | 1 | 0 | 0 | 0 |
| $F_{11}$ | 0 | 0 | 0 | 0 | 0 | 0 | 0 | 0 | 0 | 0 | 0 | 0 | 0 | 1 | 0 | 0 | 0 | 0 | 0 | 1 |
| $F_{12}$ | 0 | 0 | 0 | 0 | 0 | 0 | 0 | 0 | 0 | 0 | 0 | 0 | 0 | 0 | 0 | 0 | 0 | 0 | 0 | 0 |
| $F_{13}$ | 0 | 0 | 1 | 0 | 1 | 0 | 0 | 0 | 0 | 0 | 0 | 0 | 0 | 0 | 0 | 0 | 0 | 0 | 0 | 0 |
| $F_{14}$ | 0 | 0 | 0 | 0 | 0 | 0 | 0 | 0 | 0 | 0 | 0 | 0 | 0 | 0 | 0 | 0 | 0 | 1 | 0 | 0 |
| $F_{15}$ | 0 | 0 | 0 | 0 | 0 | 0 | 0 | 0 | 0 | 0 | 0 | 0 | 0 | 0 | 0 | 0 | 0 | 0 | 0 | 0 |
| $F_{16}$ | 0 | 0 | 0 | 0 | 0 | 1 | 0 | 0 | 0 | 0 | 0 | 1 | 0 | 0 | 0 | 0 | 0 | 0 | 0 | 0 |
| $F_{17}$ | 0 | 0 | 0 | 0 | 0 | 0 | 0 | 0 | 0 | 0 | 0 | 0 | 0 | 0 | 0 | 0 | 0 | 0 | 0 | 0 |
| $F_{18}$ | 0 | 0 | 0 | 0 | 0 | 0 | 0 | 0 | 1 | 0 | 0 | 0 | 0 | 0 | 0 | 0 | 0 | 0 | 1 | 0 |
| $F_{19}$ | 0 | 0 | 0 | 0 | 0 | 0 | 0 | 0 | 0 | 0 | 0 | 0 | 0 | 0 | 0 | 1 | 0 | 0 | 0 | 0 |
| $F_{20}$ | 0 | 0 | 0 | 0 | 0 | 0 | 0 | 0 | 0 | 0 | 1 | 0 | 0 | 1 | 0 | 0 | 0 | 0 | 0 | 0 |

**Figure 3.** Adjacency matrix A of factors influencing organizational resilience.

### 4.2.2. Calculate the Reachable Matrix

The reachable matrix is mainly used to represent the direct or indirect action relationship between the influencing factors, such as the influencing factor $F_i$ can reach $F_j$ through the distance of cell 1. Similarly, $F_j$ can reach the next influencing factor through the distance of cell 1. Therefore, according to the Boolean rule, if the adjacency matrix A satisfies the condition: $(A + I)^{k-1} \neq (A + I)^k = (A + I)^{k+1} = MC$, the obtained matrix M is the reachable matrix of the adjacency matrix A. The operation of the reachable matrix M is performed using Matlab software, and the results are shown in Figure 4 below.

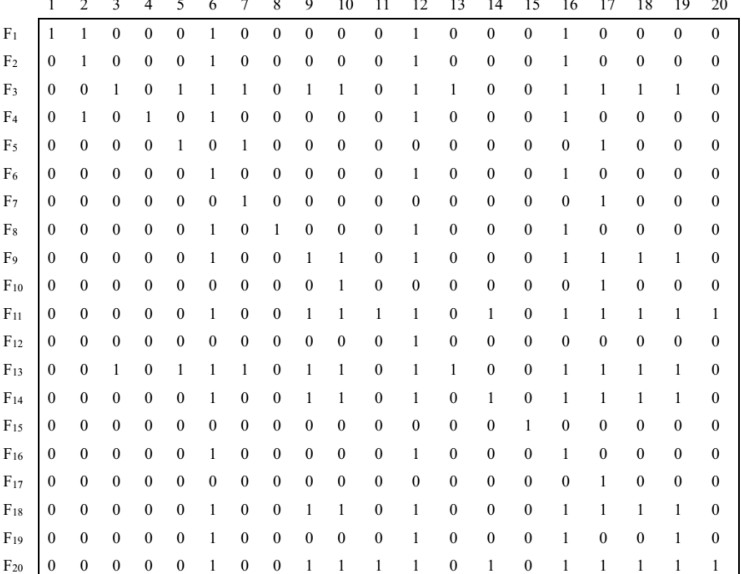

**Figure 4.** Reachable matrix M of factors influencing organizational resilience.

### 4.2.3. Hierarchical Processing of Reachable Matrices

In this study, the set $C(F_i)$ and the set $D(F_i)$ are obtained on the basis of the reachable matrix M. The set $C(F_i)$ represents the set of the elements of the reachable matrix Fi in the row containing the elements of the column corresponding to "1". The set $D(F_i)$ represents the set of the elements of the reachable matrix Fi in the column containing the elements of the row corresponding to "1". The common set $E(F_i)$ represents the set consisting of the intersection between the set $C(F_i)$ and the set $D(F_i)$, i.e., the set that can influence $C(F_i)$ and be influenced by $D(F_i)$ at the same time. The results of the reachable matrix hierarchy processing are shown in Table 4 below.

**Table 4.** Hierarchical treatment of the reachable matrix of factors influencing organizational resilience.

| $F_i$ | $C(F_i)$ | $D(F_i)$ | $E_{(F_i)} = C(F_i) \cap D(F_i)$ |
|---|---|---|---|
| $F_1$ | 1, 2, 6, 12, 16 | 1 | 1 |
| $F_2$ | 2, 6, 12, 16 | 1, 2, 4 | 2 |
| $F_3$ | 3, 5, 6, 7, 9, 10, 12, 13, 16, 17, 18, 19 | 3, 13 | 3, 13 |
| $F_4$ | 2, 4, 6, 12, 16. | 4 | 4 |
| $F_5$ | 5, 7, 17 | 3, 5, 13 | 5 |
| $F_6$ | 6, 12, 16 | 1, 2, 3, 4, 6, 8, 9, 11, 13, 14, 16, 18, 19, 20 | 6, 16 |
| $F_7$ | 7, 17 | 3, 5, 7, 13 | 7 |
| $F_8$ | 6, 8, 12, 16 | 8 | 8 |
| $F_9$ | 6, 9, 10, 12, 16, 17, 18, 19 | 3, 9, 11, 13, 14, 18, 20 | 9, 18 |
| $F_{10}$ | 10, 17 | 3, 9, 10, 11, 13, 14, 18, 20 | 10 |
| $F_{11}$ | 6, 9, 10, 11, 12, 14, 16, 17, 18, 19, 20 | 11, 20 | 11, 20 |
| $F_{12}$ | 12 | 1, 2, 3, 4, 6, 8, 9, 11, 12, 13, 14, 16, 18, 19, 20 | 12 |
| $F_{13}$ | 3, 5, 6, 7, 9, 10, 12, 13, 16, 17, 18, 19 | 3, 13 | 3, 13 |
| $F_{14}$ | 6, 9, 10, 12, 14, 16, 17, 18, 19 | 11, 14, 20 | 14 |
| $F_{15}$ | 15 | 15 | 15 |
| $F_{16}$ | 6, 11, 16 | 1, 2, 3, 4, 6, 7, 8, 9, 11, 13, 14, 16, 18, 19, 20 | 6, 11, 16 |
| $F_{17}$ | 17 | 3, 5, 7, 9, 10, 11, 13, 14, 17, 18, 20 | 17 |
| $F_{18}$ | 6, 9, 10, 12, 16, 17, 18, 19 | 3, 9, 11, 14, 15, 18, 20 | 9, 18 |
| $F_{19}$ | 6, 12, 16, 19 | 3, 9, 11, 13, 14, 18, 19, 20 | 19 |
| $F_{20}$ | 6, 9, 10, 11, 12, 14, 16, 17, 18, 19, 20 | 11, 20 | 11, 20 |

### 4.2.4. Constructing an ISM of the Factors Influencing Organizational Resilience

The hierarchical classification of influencing factors of organizational resilience is based on $E(F_i) = C(F_i) \cap D(F_i)$ to be extracted level by level. For example, after the first hierarchical process, the results that satisfy $E(F_i) = C(F_i) \cap D(F_i)$ are 12, 15, and 17, so {12, 15, and 17} is the first level. After that, the elements containing 12, 15, 17 are removed from the list and 6, 7, 10, 16 are found to satisfy the condition, so 6, 7, 10, 16 is the second layer, and so on until all the layers are found.

Using Matlab software, the final hierarchical results were obtained as follows: $L_1 = \{12, 15, 17\}$; $L_2 = \{6, 7, 10, 16\}$; $L_3 = \{2, 5, 8, 19\}$; $L_4 = \{1, 4, 9, 18\}$; $L_5 = \{3, 13, 14\}$; $L_6 = \{11, 20\}$. Based on the results of the hierarchical analysis, the reachability matrix and the original adjacency matrix, the ISM of the factors influencing organizational resilience

was constructed by converting the variable symbols into their corresponding elements, as shown in Figure 5 below.

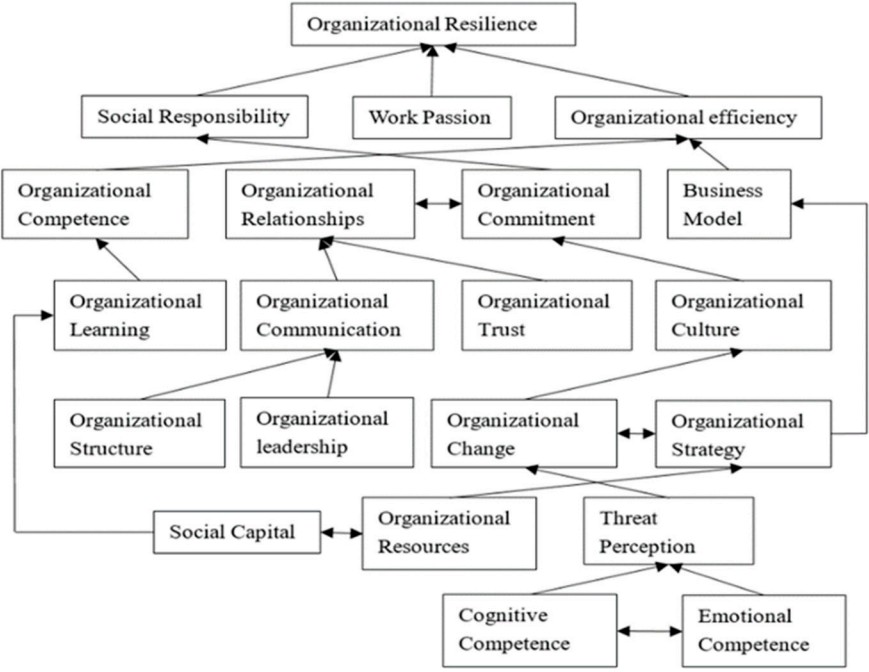

**Figure 5.** ISM of the factors influencing organizational resilience.

As can be seen from Figure 5 above, the 20 influencing factors affecting organizational resilience constitute a 6-step hierarchical model, and the influencing factors in each stratum exhibit differentiation among themselves. At first, the first level of social responsibility, work passion and organizational efficiency and the second level of organizational capability, organizational relationship, organizational commitment, and business model, are the surface-level influences on organizational resilience, which are the direct influences on organizational resilience. As a general term for corporate behavior, social responsibility can bring benefits to multiple stakeholders beyond legal requirements. Active fulfillment of CSR helps to enhance the stability of resilient organizations by improving their ability to absorb external shocks and weakening the degree of negative impact of external events on the organization. The fulfillment of social responsibility helps to gain the support of stakeholders and increase reciprocity, and the company becomes more firmly integrated into its social and natural environment. For work passion, maintaining positive emotions, such as hope and optimism, can help individuals maintain positive perceptions and help generate passionate, creative, and positive coping behaviors that induce employee resilience. Organizational efficiency helps to enhance organizational performance in a resource-consuming situation, and at the same time, increased organizational efficiency helps to equip organizational members with the capabilities needed to cope with environmental changes. Organizational capabilities and organizational relationships characterize the organizational capabilities needed to enhance organizational resilience. Organizational commitment helps to enhance the trust of stakeholders outside the organization, which can give the organization more resources and opportunities and show that the organization dares to take responsibility. Plus, it helps to enhance the efficiency of dealing with problems within the organization and achieve the result of doing what it says. Business model is also important as the mode of organization operation; a good business model helps the organization to run efficiently, achieve the expected organizational goals, and enhance the organization's ability to cope with crises. Therefore, these seven factors are the direct causes of high organizational resilience, and several other layers of factors act on organizational resilience by influencing the surface factors.

Second, the middle-level influencing factors of organizational resilience are the third and fourth levels of a total of eight factors, which have an indirect impact on organizational resilience. Among them, organizational learning, organizational communication, and organizational change belong to the category of organizational actions, which characterize the organizational actions taken to enhance organizational resilience. Organizational trust belongs to the category of organizational resources, which helps the organization to obtain a wide range of resources, enhance the organization's ability to cope with crises and obtain support and help from other organizations more easily. Organizational leadership and organizational strategy belong to the category of organizational mode, which represents the role of organizational operation mode on organizational resilience. Organizational culture and organizational structure belong to the attribute characteristics of an organization, which indicates that the inherent culture and structure of an organization are the fundamental influencing factors of organizational resilience. Therefore, the intangible culture formed by the organization and the initial structure of the organization determine the crisis the organization can withstand. These mid-level influences influence organizational resilience through their constraining effect on surface-level influences.

Third, the bottom five factors are social capital, organizational resources, threat perception, cognitive ability, and emotional ability, which are the deeper factors of organizational resilience. In other words, organizational resources, organizational members' perception of threat, organizational members' cognitive ability and organizational members' ability to control emotions do not directly affect organizational resilience, but they can affect organizational resilience through other influencing factors. This also verifies the results of the previous studies. For example, social capital will affect the work efficiency and work enthusiasm of employees to some extent by influencing the degree of coordination and cooperation in the work of employees, thus enhancing the resilience of organizations in the face of crisis [57,62].

## 5. Importance Analysis of Factors Influencing Organizational Resilience

From the ISM established in the previous section, it is clear that the influencing factors within organizational resilience are not independent but are interdependent and interacting. These characteristics limit the use of Analytic Hierarchy Process (AHP). Therefore, the Analytic Network Process (ANP) is more realistic for the study of the importance of each influencing factor. It not only retains the advantages of AHP, but also eliminates the assumption that internal elements are independent of each other and can better describe the system with complex structure and internal dependencies. However, due to the tedious manual calculation process of ANP, Super Decision (SD) software is used to calculate the weights of each influencing factor in this paper.

### *5.1. Analysis Process of ANP Model*

ANP is a decision-making method that adapts to a non-independent recursive hierarchy, which is a new practical decision-making method developed on the basis of AHP. It is particularly suitable for complex decision-making systems with internal dependency and feedback relationships [74].

### 5.1.1. Network Structure Construction of Decision Indicators

The ANP network divides the system elements into two main parts. The first part is the control factor layer, which includes the problem objective and the decision criterion. There can be no decision criterion in the control layer, but there is at least one objective. The second part is the network layer, which is a network structure formed by the elements that interact with each other.

### 5.1.2. ANP Weightless Supermatrix Construction

With the criterion in the control layer relative to the target layer be $P_1, \ldots, P_m$, the network layer has elements $C_1, \ldots, C_n$. With the control layer element $P_S$ (s = 1, 2

..., m) as the criterion and the element $C_{jl}$ (l = 1, 2 ..., $n_j$) C in $C_j$ as the sub-criterion, the influence size of each element in the element group $C_j$ on $C_{jl}$ is judged, and the judgment matrix is constructed and the normalized feature vector $\left[ W_{i1}^{j1}, W_{i2}^{j1} ..., W_{in}^{j1} \right]^T$ is obtained, which is the sorting vector of the network layer elements. Similarly, the ranking vectors relative to the other elements can be obtained, and a matrix $W_{ij}$ can be summarized, whose column vector elements $C_{i1}, ..., C_{in}$ is the importance sorting vector of the elements in $C_j$.

$$W_{ij} = \begin{bmatrix} W_{i1}^{j1} & \cdots & W_{i1}^{jn} \\ \vdots & \ddots & \vdots \\ W_{in}^{j1} & \cdots & W_{in}^{jn} \end{bmatrix}$$

If the group of elements $C_i$ not related to $C_j$ is uncorrelated, then $W_{ij} = 0$. Organizing the sorted vectors of all network layer elements interacting with each other yields a control element under $P_s$ under the unweighted supermatrix $W_s$.

### 5.1.3. Weight Supermatrix Construction

With $P_s$ as the main criterion and $C_j$ as the secondary criterion, we construct the judgment matrix $A_j$ and normalize it to find the normalized eigenvectors $\left[ a_{1j}, a_{2j}, a_{3j,...}, a_{nj} \right]^T$. Similarly, we can find the weight matrix $A_s$ for the relationships between the elements under $P_s$:

$$A_s = \begin{bmatrix} a_{11} & \cdots & a_{1n} \\ \vdots & \ddots & \vdots \\ a_{n1} & \cdots & a_{nn} \end{bmatrix}$$

Take the weight matrix $A_s$ times the unweighted supermatrix $W_S^w$:

$$W_s^w = A_s W_s$$

### 5.1.4. Limit Supermatrix Solution

In the ANP method, in order to reflect the dependencies between the elements, it is necessary to perform a stabilization of the weighted supermatrix $W_S^w$ to form a stabilization process, i.e., to calculate the limit relative ranking vector $W_S^1$. Its column j is the limit relative ranking of each element in the network layer with respect to element j, i.e., the weight value of each element with respect to the highest target:

$$W_S^1 = \lim_{k \to +\infty} W^k$$

where $W_S^1$ denotes the limit supermatrix and $W$ is the total weight supermatrix.

### 5.2. Establishment of ANP for Factors Influencing Tissue Toughness

By consulting with relevant experts, the inter-influence relationship between organizational resilience influencing factors was determined. Then, the ANP model of organizational resilience influencing factors with dependency relationship within the network was established (see Figure 6). The control layer of this structural model has only the target, i.e., organizational resilience influencing factors. The network layer includes five groups of elements that influence organizational resilience. Among them, organizational action factor ($C_1$) includes organizational communication ($C_{11}$), organizational learning ($C_{12}$), organizational commitment ($C_{13}$), organizational change ($C_{14}$), and organizational efficiency ($C_{15}$). Organizational model factor ($C_2$) includes organizational strategy ($C_{21}$), business model ($C_{22}$), and organizational leadership ($C_{23}$). Organizational attribute factor ($C_3$) includes organizational structure ($C_{31}$), organizational culture ($C_{32}$), and social responsibility ($C_{33}$). Organizational capacity factor ($C_4$) includes organizational competence ($C_{41}$), emotional competence ($C_{42}$), cognitive competence ($C_{43}$), threat perception ($C_{44}$) and work passion

($C_{45}$). Organizational resource factor ($C_5$) includes organizational resource ($C_{51}$), organizational relationship ($C_{52}$), social capital ($C_{53}$) and organizational trust ($C_{54}$). The identified indicators and their relationships are entered into the SD software to form a network structure diagram. The circular arrows indicate the existence of inter-influence relationships within indicators and the direct arrows indicate the existence of inter-influence relationships between indicator groups.

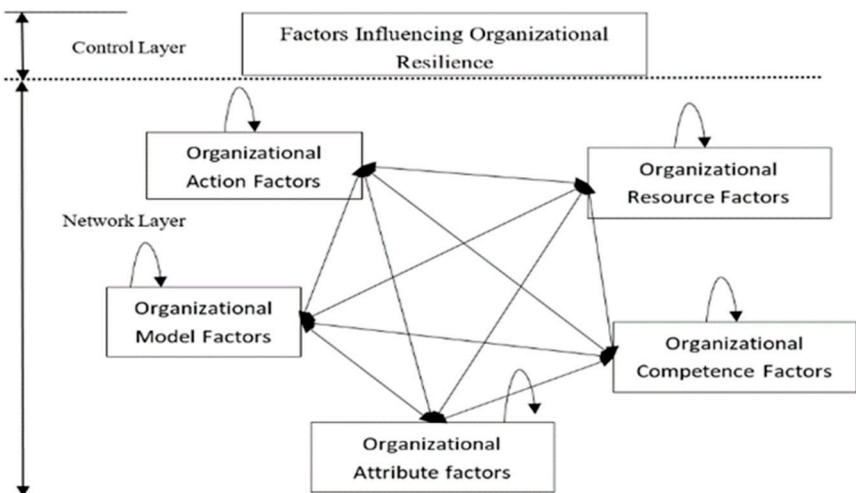

**Figure 6.** ANP model diagram of factors influencing organizational resilience.

### 5.3. Calculation of the Weights of Factors Influencing Organizational Resilience

#### 5.3.1. Construction of Weightless Supermatrix

From the diagram of the ANP model of factors influencing organizational resilience, we can see that there is only one element in the control layer, so the element in the control layer is denoted as $P_1$. There are five element groups in the network layer, denoted as $C_1$, $C_2$, $C_3$, $C_4$, $C_5$. Let the elements contained in each element group $C_i$ ($i = 1, 2, ..., 5$) be $C_{i1}$, $C_{i2}$, ..., $C_{iN}$, with $P_1$ as the main criterion and $C_{ik}$ ($k = 1, 2, ..., N_i$) as the sub-criterion for the elements in $C_i$. Compare elements in each element group in pairs according to their influence on $C_{ik}$ to obtain the judgment matrix. By combining the normalized eigenvectors of each judgment matrix, we can find the unweighted hypermatrix $W$.

The experts in the consulting group were consulted and the nine-degree method was used to input the obtained judgments into the Super Decision software, and the unweighted super matrix $W$ was calculated.

#### 5.3.2. Build the Weight Supermatrix

With $P_1$ as the main criterion and the element group $C_i$ ($i = 1, 2, 3, 4, 5$) as the sub-criterion, pairwise comparisons of other element groups are made to obtain the judgment matrix. Plus, the normalized eigenvectors of the judgment matrix are combined to obtain the weighting matrix $A$. The weight supermatrix can be expressed as: $W = AW$.

#### 5.3.3. Limit Supermatrix Solution

After iterative processing of the weighted hypermatrix, the limit hypermatrix can be obtained: $W_s : W_s = \lim\limits_{R \to \infty} (W)^k$. If the limit supermatrix $W_s$ converges and is unique, each row of data in the matrix is the same, and each column is the weight corresponding to each element. Using Super Decision software, the limit supermatrix is solved, and the weights and ranking of each index are shown in Table 5.

**Table 5.** Weighting and ranking of factors influencing organizational resilience.

| Factors | Weights | Sort by |
|---|---|---|
| $C_{11}$ Organizational Communication | 0.197 | 4 |
| $C_{12}$ Organizational Learning | 0.167 | 7 |
| $C_{13}$ Organizational Commitment | 0.027 | 17 |
| $C_{14}$ Organizational Change | 0.024 | 18 |
| $C_{15}$ Organizational Efficiency | 0.033 | 15 |
| $C_{21}$ Organizational Strategy | 0.168 | 6 |
| $C_{22}$ Business Model | 0.076 | 9 |
| $C_{23}$ Organizational Leadership | 0.072 | 10 |
| $C_{31}$ Organizational Structure | 0.007 | 20 |
| $C_{32}$ Organizational Culture | 0.031 | 16 |
| $C_{33}$ Social Responsibility | 0.018 | 19 |
| $C_{41}$ Organizational Competence | 0.214 | 2 |
| $C_{42}$ Emotional Competence | 0.053 | 14 |
| $C_{43}$ Cognitive Competence | 0.053 | 13 |
| $C_{44}$ Threat Perception | 0.067 | 12 |
| $C_{45}$ Work Passion | 0.080 | 8 |
| $C_{51}$ Organizational Resources | 0.227 | 1 |
| $C_{52}$ Organizational Relationships | 0.213 | 3 |
| $C_{53}$ Social Capital | 0.170 | 5 |
| $C_{54}$ Organizational Trust | 0.068 | 11 |

*5.4. Analysis of Results*

According to the weighted ranking of the influencing factors in Table 5, it can be seen that the main factors affecting organizational resilience are: organizational resources, organizational competence, organizational relationships, organizational communication, social capital, organizational strategy, organizational learning, and work passion. Therefore, although there are a variety of factors affecting organizational resilience, as long as the key ones are grasped, they can play a great supporting role in improving organizational resilience. At the same time, it is crucial to improve the organization's ability to deal with crisis by providing necessary resources and improving the organization's ability to deal with crisis. For example, resources can be fully utilized and effectively replaced through resource combination and advanced experience of other organizations can be converted into their own knowledge and ability.

**6. Conclusions and Outlook**

*6.1. Research Conclusions*

Profitability from turbulent environments, sustainable development and competitive advantage are issues that organizations now face and need to address. Organizational resilience is the necessary ability for an organization to resist the interference of various risks and realize survival and development in a turbulent and changing environment. Exploring the influencing factors of organizational resilience can help organizations better understand how to deal with the crisis and guide the path to improve organizational resilience. In this study, five companies are taken as the object of multi-case study, and the influencing factors of organizational resilience are extracted on the basis of multi-case analysis. The explanatory structure model of the influencing factors of organizational

resilience is constructed by using ISM method, and the importance of the influencing factors of organizational resilience is analyzed by using ANP method.

First, by exploring the influencing factors of organizational resilience, this study finds that the influencing factors of organizational resilience can be systematically summarized at three levels: surface level, middle level, and deep level. Among them, the surface-level influencing factors refer to which influencing factors directly affect organizational resilience, mainly including organizational competence, organizational relationships, organizational learning, and organizational communication. Middle level influencing factors refer to the factors that need to be reflected by deep-seated inquiry into organizational behavior, mainly including organizational culture, organizational structure and organizational leadership. Deep influence factors refer to the factors that affect organizational resilience at the deepest level, mainly including social capital, organizational resources, cognitive competence, and emotional competence. Subsequently, this study selected Southwest Airlines, Apple, Microsoft, Starbucks, and Kyocera as the research objects through multi-case analysis. The influencing factors of organizational resilience were found to include five aspects of organizational action factors, organizational model factors, organizational attribute factors, organizational competence factors, and organizational resource factors, with a total of 20 influencing factors. Meanwhile, in addition to the influencing factors identified by scholars, this study finds that the influencing factors of organizational resilience include social responsibility, work passion, organizational efficiency, organizational commitment, business model, and organizational strategy through a multi-case analysis.

Second, this study explored the influencing factors of organizational resilience using ISM and constructed an ISM of influencing factors of organizational resilience. It is found that the 20 influencing factors of organizational resilience constituted a 6-step hierarchical model, and the influencing factors of each stratum showed differentiation among themselves. The surface-level influences on organizational resilience include social responsibility, work passion, and organizational efficiency, which are direct influences on organizational resilience. The middle-level influencing factors of organizational resilience include 12 influencing factors, such as organizational competence, organizational relationships and organizational trust, which have indirect influence on organizational resilience. The bottom five factors are social capital, organizational resources, threat perception, cognitive competence, and emotional competence, which are the deeper factors of organizational resilience, and they can influence organizational resilience by acting on other influencing factors. Subsequently, this study used ANP to investigate the importance of organizational resilience factors and found that the main factors affecting organizational resilience are organizational resources, organizational competence, organizational relationships, organizational communication, social capital, organizational strategy, organizational learning, and work passion.

*6.2. Discussion and Practical Insights*

Compared with existing studies on organizational resilience, this study adopts the multi-case study method to explore the influencing factors of organizational resilience by taking five high-resilience enterprises as the research objects. An ISM of the influencing factors of organizational toughness was established, and the importance of the influencing factors was analyzed by AHP. This study is not only a supplement to existing research, but also a supplement and innovation on the basis of existing research. First, existing studies have found that organizational capacity [33], organizational relationships [56], and organizational learning [30,31] are influential factors affecting organizational resilience. This study extracted these influences and validated the findings based on a multi-case analysis. Furthermore, this study found that the influencing factors of organizational resilience include social responsibility, work passion, organizational efficiency, organizational commitment, business model, and organizational strategy, which are useful additions to the existing studies.

In addition, this study classifies the variables into superficial influencing factors, middle influencing factors and deep influencing factors based on their influence relationship on organizational resilience. In this study, the hierarchical relationship of the influencing factors of organizational toughness was verified by constructing an ISM. On the other hand, this study clarified the relationship among influencing factors through the ISM of the influencing factors of organizational toughness, which is different and related to the existing conclusions. Among them, established studies point out that organizational culture [15,28], organizational structure, organizational leadership [58], are middle and deep influencing factors of organizational resilience. The findings of this study support this view. Meanwhile, the difference with the results of the present study is that scholars, such as Valero et al. (2015) [33], Gittell et al. (2006) [9], and Mithani et al. (2020) [31], stated that organizational learning and organizational communication are direct influences on organizational resilience. However, this study found that organizational learning and organizational communication are indirect influencing factors of organizational resilience, which may be caused by the fact that existing studies have not fully discovered and opened up the relationship between the influencing factors of organizational resilience. In fact, it is not difficult to understand from the ISM of organizational resilience and real organizational practices that organizational learning helps organizations to absorb knowledge and transform it into competencies, while the ability to learn from crises improves organizational resilience [75]. Nathan and Kovoor-Misra (2002) [76] also noted the importance of learning from other organizations for crisis management. They highlight that companies that build capital by sharing knowledge and learning from each other will be able to better minimize outside disruption to the organization. In times of organizational strife and disruption, the frustration and anxiety felt by organizational members also increases [77], and frustration and anxiety in turn may lead to collective employee turnover [78]. Regardless of the origin of organizational member withdrawal, lesser organizational interactions can stifle the social networks that sustain the organization. This is possibly because critical information about internal activities cannot be shared with others in the organization [77]. Therefore, good organizational communication helps to restore and strengthen organizational relationships, reshape the organization's social network, and enable knowledge sharing and exchange with other members of the organization, thereby enhancing the organization's ability to cope with crises and increase organizational resilience.

This study finds that the main factors influencing organizational resilience are organizational resources, organizational capabilities, and organizational relationships through the analysis of the importance of the factors influencing organizational resilience. Further, organizational resources and organizational capabilities are the two most important factors influencing organizational resilience, which coincides with the resource-capability doctrine in existing research. The organizational capability view assumes that the competitive advantage of a firm exists in the resource structure. In a dynamic and complex environment, firms build and enhance their ability to cope with crises and maintain competitive advantage by combining resources and establishing new resource structures to cope with changes outside the organization [79]. The resource selection mechanism influences the firm's output process in terms of resource acquisition decisions, while the capability building mechanism influences the firm's output in various processes and aspects of resource allocation, which together constitute the completion process of production within the firm. Thus, the integration based on resources and capabilities forms a firm's competitive strategy that enhances organizational resilience and maintains a unique competitive advantage even when the firm is in an environment of technological and demand changes.

The practical insight of this study is that shaping organizational resilience and accumulating corporate resilience assets does not happen overnight, but requires both time and a long-term strategic design, detailed plans, and effective measures. Specifically, first enhancing organizational resilience is key. Resilience can help organizations respond and adapt to a crisis, thus affecting the speed at which an organization can respond to a crisis. First, we need to strengthen organizational cohesion. Only by complementing each other's

abilities and making joint efforts can teams show stronger cohesion and anti-pressure ability. Good communication between members of an organization is conducive to effective coordination between members and organizations, so as to change strategies and solve problems in real time in response to crises. Good communication contributes to the formation of a knowledge-sharing atmosphere in the organization, making common progress through mutual learning, thus enhancing resilience. Training and education should be provided to organization members. Effective training and education will help organization members to take the initiative to learn and transform the knowledge learned into ability. At the same time, the development of resilience at the individual level, team level and organizational level should be strengthened.

Second, organizational learning is an important foundation. First, organization members should establish a correct view of crisis. Crisis exists in the dynamic environment everywhere in an organization, and the crisis is regarded as the organization's ability to achieve "corner overtaking" and the opportunity to achieve competitive advantage. Second, pay attention to the alternation of learning modes, such as progressive learning, double-loop learning and single-loop learning. Organization members should identify the characteristics of different learning modes and the best applicable scenarios, and correctly use the learning mode to achieve twice the result with half the effort. Third, to become a learning organization, the crisis of the enterprise needs to be faced and solved by the members of the enterprise, not just by the responsibility of the management. Therefore, to form a learning team before the emergence of major problems, learning from each other and solving problems together is essential.

Third, emotion management is a necessity. Emotions not only affect the ability of organization members to predict the occurrence of crises, but also affect the speed of crisis detection, as well as the effectiveness of organizational actions due to changes in their emotions. Organization members should make an objective analysis of the crisis, and on this basis, strengthen the positive self-psychological suggestion by means of self-dialogue and communication with organization members. They should turn their attention to other good things to replenish their mental resources and improve their personal resilience.

Fourth, organizational resources and basic security. In the process of organizational development, having sufficient organizational resources is more conducive to the organization to take organizational actions quickly and seize market opportunities. One is to increase the possibility of discovering market opportunities through interaction with people in the organization's social network. Using social networking platforms to introduce the organization to key customers, partners, etc., thereby increases the likelihood of organizational action. Secondly, through the network relations of organization members, they can obtain resources that they do not have, expand the resource pool, and realize the maximum and effective utilization of resources through resource patchwork.

*6.3. Research Shortcomings and Outlook*

However, as an exploratory study, the shortcomings of this study are: first, organizational resilience has industry differences, and organizational resilience in different industries may be influenced by different influencing factors. The sample of this study is mainly from the passenger aircraft industry, and the characteristics of organizational resilience and the influencing factors of other industries need to be further developed and explored. Second, although this study analyzed the influencing factors of organizational resilience and explored the relationship between the influencing factors and their structure in depth, it did not further investigate the relationship between the influencing factors and the effect on organizational toughness by means of empirical research. The logical relationships among the influencing factors can be verified by means of empirical studies in the future.

**Author Contributions:** Conceptualization, R.C. and Y.L.; methodology, F.Z.; validation, Y.L. and R.C.; formal analysis, S.Z.; investigation, J.W.; resources, R.C. and S.Z.; data curation, R.C.; writing—

original draft preparation, R.C.; writing—review and editing, Y.L.; visualization, S.Z.; supervision, Y.L. and R.C. All authors have read and agreed to the published version of the manuscript.

**Funding:** This research was funded by the Fundamental Research Funds for the Central Universities (2021YJS048).

**Institutional Review Board Statement:** Not applicable.

**Informed Consent Statement:** Not applicable.

**Data Availability Statement:** Not applicable.

**Acknowledgments:** The authors would like to thank the anonymous reviewers for their reviews and comments.

**Conflicts of Interest:** The authors declare no conflict of interest.

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
