# Peer review of "Analysis of the Influencing Factors of Organizational Resilience in the ISM Framework: An Exploratory Study Based on Multiple Cases"

_sustainability, doi:10.3390/su132313492_

Round 1

Reviewer 1 Report

In relation to present theoretical background and empirical research it is worth to:

  • analyze and include (maybe close to lines 47-51) main disciplinary perspectives represented in the organizational resilience literature. It is worth to see an article presenting the results of a systematic review of an extensive literature: Hillmann, J. Disciplines of organizational resilience: contributions, critiques, and future research avenues. Review of Managerial Science 15, 879–936 (2021). https://doi.org/10.1007/s11846-020-00384-2
  • see an article with literature review of empirical research on organizational resilience which was conducted to summarize the diverse findings of 69 studies, focusing on the factors that lead to resilience: Vakilzadeh, K. and Haase, A. (2021), "The building blocks of organizational resilience: a review of the empirical literature", Continuity & Resilience Review, Vol. 3 No. 1, pp. 1-21. https://doi.org/10.1108/CRR-04-2020-0002

Conclusions are thoroughly supported by the results presented in the article. In area concerned training, education, organizational learning and emotion management (lines 713-737) it is worth to underline that “The employee is the basic elements of organization system and the individual resilience is the main source of organizational resilience.” (Lei XIAO, Huan CAO, Organizational Resilience: The Theoretical Model and Research Implication, ITM Web of Conferences 12, 04021 (2017), DOI: 10.1051/itmconf/20171204021) “Individual has resilience does not mean that groups or organizations he or she belongs to also have such characteristics” – this sentence relates to the content presented in article and guides the reader to practical tips on what to do in building the OR (maybe it is worth to expand conclusions for example, the three levels of OR: individual, group and organizational).

“Therefore, today more than ever, companies need to pay extra attention to fostering organizational resilience [16], and tapping into the core influences of organizational resilience is the key to finding out how organizations can achieve sustained competitive advantage” (lines 43-46). In connection with this introduction, it is worth extending the summary with tips on what actions to take to strengthen OR. For example Business Continuity Management (BCM) which concentrates on key product(s) as a technical and organizational side of building OR.

Table 3, level F11 (emotional competence) – font should be modified

I find the article very interesting, systematizing the organizational resilience issues and clearly presenting the results that can be used in management practice.

Author Response

Comment 1: analyze and include (maybe close to lines 47-51) main disciplinary perspectives represented in the organizational resilience literature. It is worth to see an article presenting the results of a systematic review of an extensive literature: Hillmann, J. Disciplines of organizational resilience: contributions, critiques, and future research avenues. Review of Managerial Science 15, 879–936 (2021). https://doi.org/10.1007/s11846-020-00384-2

Response: The author has read the article “Hillmann, J. Disciplines of organizational resilience: contributions, critiques, and future research avenues. Review of Managerial Science 15, 879–936 (2021)” carefully and added relevant content to the article.

Comment 2: see an article with literature review of empirical research on organizational resilience which was conducted to summarize the diverse findings of 69 studies, focusing on the factors that lead to resilience: Vakilzadeh, K. and Haase, A. (2021), "The building blocks of organizational resilience: a review of the empirical literature", Continuity & Resilience Review, Vol. 3 No. 1, pp. 1-21. https://doi.org/10.1108/CRR-04-2020-0002

Response: The authors summarize the article by Vakilzadeh, K. and Haase, A. (2021) and present it in the section "2.2 Factors influencing organizational resilience". The details are as follows:

From the research within the study, only scholars Vakilzadeh & Haase (2021)[53] summarized and explored the influencing factors of organizational resilience based on the empirical study of organizational resilience, however, it is difficult to reflect the cross-level characteristics of organizational resilience. In view of this, this study sys-tematically sorts out the influencing factors of organizational resilience from three lev-els, namely surface, middle and deep levels, based on the previous studies, in order to more clearly reflect the characteristics of the influencing factors of organizational re-silience.

Comment 3: Conclusions are thoroughly supported by the results presented in the article. In area concerned training, education, organizational learning and emotion management (lines 713-737) it is worth to underline that “The employee is the basic elements of organization system and the individual resilience is the main source of organizational resilience.” (Lei XIAO, Huan CAO, Organizational Resilience: The Theoretical Model and Research Implication, ITM Web of Conferences 12, 04021 (2017), DOI: 10.1051/itmconf/20171204021) “Individual has resilience does not mean that groups or organizations he or she belongs to also have such characteristics” – this sentence relates to the content presented in article and guides the reader to practical tips on what to do in building the OR (maybe it is worth to expand conclusions for example, the three levels of OR: individual, group and organizational).

Response: The authors have revised the revelation section of the paper by examining it, in which the authors state that, at the same time, the development of resilience at the individual level, the team level, and the organizational level should be enhanced.

Comment 4: Therefore, today more than ever, companies need to pay extra attention to fostering organizational resilience [16], and tapping into the core influences of organizational resilience is the key to finding out how organizations can achieve sustained competitive advantage” (lines 43-46). In connection with this introduction, it is worth extending the summary with tips on what actions to take to strengthen OR. For example Business Continuity Management (BCM) which concentrates on key product(s) as a technical and organizational side of building OR.

Response: We thank the experts for their comments, and the authors have revised the Conclusions and Implications section accordingly.

Comment 5: Table 3, level F11 (emotional competence) – font should be modified

Response: The authors have made changes to the format.

Reviewer 2 Report

The paper is very general in the way it tackles organizational resilience. It takes into consideration several influencing factors without a specific focus on any of these factors. I disagree with the authors that the issue of organizational resilience is under explored, actually the literature is full with studies on resilience. What is expected next is to pay more attention on specific issues that need further exploration rather than repeating the same concepts. This is not to say that the paper has nothing new to offer, but rather what could be made/added is another section that theoretically supports the findings of this research. For instance, if organizational commitment was found to be influential, then please state who theoretically supported this point from the literature previously. Also, refer to: 

Sawalha, I. (2015). ‘Managing Adversity: Understanding some Dimensions of Organizational Resilience’, Management Research Review, Vol. 38, No. 4, pp. 346-366, Emerald.

This article provides an excellent insight into organizational resilience and the factors influencing resilience. Also, it would be useful to summarize the factors influencing organizational resilience from a social (organizational) perspective and those influencing resilience from a systems perspective since the authors are dealing with the subject from a more technical perspective. 

Author Response

Comment 1: The paper is very general in the way it tackles organizational resilience. It takes into consideration several influencing factors without a specific focus on any of these factors. I disagree with the authors that the issue of organizational resilience is under explored, actually the literature is full with studies on resilience. What is expected next is to pay more attention on specific issues that need further exploration rather than repeating the same concepts. This is not to say that the paper has nothing new to offer, but rather what could be made/added is another section that theoretically supports the findings of this research. For instance, if organizational commitment was found to be influential, then please state who theoretically supported this point from the literature previously. Also, refer to:

Sawalha, I. (2015). ‘Managing Adversity: Understanding some Dimensions of Organizational Resilience’, Management Research Review, Vol. 38, No. 4, pp. 346-366, Emerald.

Response: The authors have revised the relevant expressions and discussion in the introduction section. The results of existing studies are also described in the discussion section and discussed with the results of this study.

Comment 2: This article provides an excellent insight into organizational resilience and the factors influencing resilience. Also, it would be useful to summarize the factors influencing organizational resilience from a social (organizational) perspective and those influencing resilience from a systems perspective since the authors are dealing with the subject from a more technical perspective.

Response: In the future research outlook section, the authors suggest that the logical relationships among the influencing factors can be further verified by means of empirical studies in the future.